# Implementation of the South African Triage Scale (SATS) in a New Ambulance System in Beira, Mozambique: A Retrospective Observational Study

**DOI:** 10.3390/ijerph191610298

**Published:** 2022-08-18

**Authors:** Andrea Conti, Daniela Sacchetto, Giovanni Putoto, Marcello Mazzotta, Giovanna De Meneghi, Emanuela De Vivo, Lorenzo Lora Ronco, Ives Hubloue, Francesco Della Corte, Francesco Barone-Adesi, Luca Ragazzoni, Marta Caviglia

**Affiliations:** 1CRIMEDIM—Center for Research and Training in Disaster Medicine, Humanitarian Aid, and Global Health, Università del Piemonte Orientale, 28100 Novara, Italy; 2Department of Translational Medicine, Università del Piemonte Orientale, 28100 Novara, Italy; 3Disaster Medicine Service 118, ASL CN1, Levaldigi, 12038 Cuneo, Italy; 4Doctors with Africa CUAMM, 35121 Padua, Italy; 5Research Group on Emergency and Disaster Medicine, Vrije Universiteit Brussels, 1050 Brussels, Belgium; 6Department of Sustainable Development and Ecological Transition, Università del Piemonte Orientale, 13100 Vercelli, Italy

**Keywords:** emergency medical service, triage, prehospital care, low- and middle-income country, South African Triage Scale

## Abstract

In 2019, an urban ambulance system was deployed in the city of Beira, Mozambique to refer patients from peripheral health centres (HCs) to the only hospital of the city (Beira Central Hospital—HCB). Initially, the system worked following a first-in–first-out approach, thus leading to referrals not based on severity condition. With the aim of improving the process, the South African Triage Scale (SATS) has been subsequently introduced in three HCs. In this study, we assessed the impact of SATS implementation on the selection process and the accuracy of triage performed by nurses. We assessed 552 and 1608 referral charts from before and after SATS implementation, respectively, and we retrospectively calculated codes. We compared the expected referred patients’ codes from the two phases, and nurse-assigned codes to the expected ones. The proportion of referred orange and red codes significantly increased (+12.2% and +12.9%) while the proportion of green and yellow codes decreased (−18.7% and −5.8%). The overall rates of accuracy, and under- and overtriage were 34.2%, 36.3%, and 29.5%, respectively. The implementation of SATS modified the pattern of referred patients and increased the number of severe cases receiving advanced medical care at HCB. While nurses’ accuracy improved with the routine use of the protocol, the observed rates of incorrect triage suggest that further research is needed to identify factors affecting SATS application in this setting.

## 1. Introduction

In low- and middle-income countries (LMICs), formal emergency medical services (EMSs) rarely exist and, when present, are often hampered by structural inefficiencies, such as poor transportation and infrastructure, the shortage of skilled medical staff and resources, and a lack of comprehensive referral protocols [1,2]. To address the transportation gap, many nongovernmental organisations (NGOs) working in LMICs have implemented EMSs at the local or regional level using different means of transport, including even motorbike and bicycle ambulances [3,4,5]. Nonetheless, the absence of well-established protocols for patient transfer frequently leads to the inappropriate transport of uncomplicated cases to referral hospitals, overwhelming their personnel and impacting already scarce available resources [6,7]. In this scenario, the implementation of patient prioritisation strategies is pivotal in managing the challenges associated with the existing mismatch between the demand for emergency care and the available resources, thus avoiding the inappropriate use of national referral hospitals and all the consequences associated with it. While prehospital triage criteria for patient transport are frequently adopted in upper-middle- and high-income countries [8], there is a lack of the scientific literature on the implementation of triage systems in LMICs’ prehospital settings, and no consensus on which triage tool works better in these contexts exists [9,10]. More broadly, only recently have LMICs started incorporating triage tools into emergency department practice by either adapting existing protocols or implementing new tools such as the South African Triage Scale (SATS) [11,12,13].

SATS, a triage protocol expressly designed for LMICs, is a four-level triage algorithm based on a list of emergency conditions and on the evaluation of seven different vital signs (mobility, respiratory rate, heart rate, systolic blood pressure, temperature, neurologic status, and history of trauma). Using three different tables (for adults, younger, and older children) containing reference values, the so-called Triage Early Warning Score (TEWS) is calculated. Patients who present one of the predetermined emergency conditions listed in the SATS protocol (obstructed airway, facial burns and inhalation, severe hypoglycaemia, and cardiac arrest) are immediately classified within the maximal priority category (red), while others have codes assigned according to the TEWS [11]. The use of objective vital-sign data renders the SATS a robust, simple, and rapid tool to be taught to inexperienced staff [14,15]. While the use of SATS has been extensively validated in different emergency departments located in resource-constrained settings, effectively reducing patient waiting time, hospital length of stay, and mortality [15,16,17,18,19,20,21], its performance in the prehospital field and in the assessment of nontrauma cases has not been thoroughly evaluated. Indeed, the formal assessment of the SATS utilisation by prehospital providers has been performed only through indirect methodologies, such as written clinical vignettes or focus-group discussions [22,23,24].

SATS has been recently integrated in the urban ambulance system of Beira, the second largest city of Mozambique with more than 530,000 inhabitants. The ambulance system was established immediately after the 2019 Cyclone Idai by NGO doctors with Africa CUAMM (Padova, Italy) in collaboration with the Centre for Research and Training in Disaster Medicine, Humanitarian Aid, and Global Health (CRIMEDIM, Università del Piemonte Orientale, Novara, Italy), with the aim of reducing hospital care referral time. The 24/7 free-of-charge referral service started its activities on 1 June 2019, linking Beira Central Hospital (HCB) with 15 peripheral health centres (HCs) through a fleet of five ambulances stationed in five different HCs, selected according to their geographical position and to the number of patients treated per day (Figure 1).

Upon its inception, the service did not contemplate a standardised prioritisation protocol for patient referral, but rather followed a first-in–first-out approach, thus often leading to a saturation of the service’s referral capacity. Therefore, SATS has been incorporated to improve referrals and regulate the number of severe cases accessing HCB from the three HCs of Chingussura, Munhava, and Manga Mascarenhas. From 27 to 30 November 2019, 75 nurses working in the above-mentioned HCs were trained on the use of SATS, and instructed to refer patients when deemed necessary according to the priority codes. Specifically, nurses underwent a single two-day course (Table A1) and were instructed to follow a specific protocol for patient transport from the three HCs to HCB (Figure A1). This is the first example of SATS integration and assessment within the referral protocol of an urban ambulance system in a low-income country. The purpose of this study was to assess whether the implementation of the SATS varied the pattern of acute patients referred from HCs to HCB. In addition, we evaluated the correct use of the SATS performed by nurses in the HCs, defined as the accuracy related to the application of the tool.

## 2. Materials and Methods

### 2.1. Study Design

This was a retrospective observational study that included all the referral transports performed from 8 October 2019 to 1 June 2020 from the three HCs of Chingussura, Manga Mascarenhas, and Munhava to HCB. The study consisted of two consecutive phases: a preimplementation phase (from 8 October to 7 December 2019) during which the assessment of patients and their subsequent referral to the HCB did not involve the use of SATS, and a postimplementation phase (from 8 December 2019 to 1 June 2020) involving the use of SATS.

### 2.2. Study Setting

The Chingussura and Munhava HCs (35 and 31 staffed beds, respectively) are classified as urban Type A health centres and were designed to serve a population of 40,000–100,000 inhabitants within a 1–4 km range. They are located at 18 and 6.5 km from HCB, respectively, and each one is equipped with one ambulance immediately available for referrals. The HC of Manga Mascarenhas (19 staffed beds) is classified as an urban Type B health centre, assisting 18,000–40,000 inhabitants within a 2–4 km range. It is 9.5 km from HCB and served by one ambulance stationed in the HC of Macurungo (at 8 km away), available upon request via phone call. Both types of urban HCs were designed to provide basic ambulatory services eventually followed by a short observational stay, such as general adult and paediatric examination, basic laboratory tests, and drug administration, while they cannot provide long-term patient hospitalisation, specialised consultation, surgery procedures, or advanced emergency care [25]. In the postimplementation phase, nurses in the three HCs were instructed to refer to HCB patients triaged with yellow, orange, and red codes, while minor green codes had to be treated at the HC level (Figure A1).

### 2.3. Data Collection

We used data from an electronic database storing all information recorded by trained nurses during transport using a patient’s referral chart (Figure A2). After collecting patients’ referral charts at HCB, dedicated local personnel trained by CUAMM performed data entry in this electronic database. Data accuracy was monitored through weekly inspections by CUAMM supervisors. Variables extracted for the present analysis were sex, age, disease category (medical, surgical/trauma, paediatric, obstetric/gynaecological), nurse-assigned priority codes (green, yellow, orange, red), and vital signs. Data were anonymised, and incomplete records for which it was not possible to retrospectively calculate the expected triage code (e.g., with one or more vital signs not recorded, or with missing age) were excluded from the statistical analysis.

### 2.4. Statistical Analysis

The researchers retrospectively assessed the expected triage code for all patients transported during the pre- and postimplementation phases using the SATS protocol. To evaluate variation in the pattern of referred acute patients, we compared the expected codes of patients transported to HCB before and after the implementation of SATS. The chi-squared test was used to evaluate changes in the proportion of the different expected triage codes.

The correct use of the SATS tool was assessed by comparing nurse-assigned and expected codes, the latter being used as the gold standard. We defined under- and overtriage as nurse-assigned codes lower or higher than the gold standard, respectively. The proportion of incorrect triage was defined as the sum of the observed proportions of under- and overtriage. To assess the inter-rater agreement, we used the quadratic weighted Fleiss’ kappa, since it was used in previous triage studies [21].

We used logistic regression to investigate the possible association between nurses’ accuracy of each referral, coded as a dichotomous variable (matching or not matching the expected code) and gender, age, and time since the implementation of SATS (independent variables). In all statistical analyses, a *p*-value of 0.05 or less was deemed to be statistically significant. We performed the analysis using Stata 15 (StataCorp, 2017, College Station, TX, USA).

## 3. Results

During the observation period, a total of 2636 referral charts were collected, and 159 and 317 records (from the pre- and postimplementation phases, respectively) were excluded. Table 1 summarises the demographic information of the considered patients. Our analysis comprised a total of 552 and 1608 referral charts for the pre- and postimplementation phases, respectively. Referral rate did not change appreciably over time. In both phases, referred patients were mainly women, and most of the referrals regarded obstetric or gynaecological and paediatric complaints (Table 1).

After the implementation of SATS, there was an increase in orange and red codes (+12.2%; +12.9%, respectively), and a reduction in green and yellow codes (−18.7%; −5.8%, respectively) referred to the HCB (Table 2). Figure 2 shows how this phenomenon took place gradually and lasted for the whole study period.

Table 3 shows the accuracy of SATS codes assigned by nurses during the postimplementation phase. The overall rate of nurses’ accuracy, and under- and over-triage were 34.2%, 36.3% and 29.5% respectively. Table A2 and Table A3 report the accuracy for obstetric or gynaecological and paediatric cases subgroups. The overall inter-rater agreement was 0.25 (95% confidence interval (CI) 0.21–0.29); in obstetric or gynaecological and paediatric cases, it was 0.17 (95% CI 0.10–0.25) and 0.16 (95% CI 0.06–0.25), respectively.

Figure 3 reports the association between time and accuracy, displaying an increasing trend from December 2019 (29.4%) to May 2020 (42.0%). This result was confirmed in the logistic regression, showing a statistically significant association between time since implementation and accuracy in the application of the tool (*p*-value < 0.0005). In contrast, the gender and age of patients did not influence nurses’ accuracy in applying the SATS protocol.

## 4. Discussion

The integration of SATS in the urban ambulance system of Beira modified the pattern of transported patients, improving the selection process and enabling local staff to become acquainted with the notions of triage and referral protocols, concepts that are still uncommon in Mozambique [23].

The increase in the proportion of severe transported cases followed a continuous progressive trend, suggesting that the concept of patient prioritisation and the SATS protocol might need time to be fully mastered by local professionals. Gradually, the first-in–first-out approach that aimed purely at reducing the pressure on constantly overcrowded HCs was replaced by a more coordinated system that has the utmost goal of addressing the limited resources available in postcyclone Mozambique to the most severe cases, optimising the use of the few ambulances, healthcare staff, and hospital assets [26]. The daily transport rates were similar before and after the SATS implementation, suggesting that the service worked from the beginning at its maximal capacity, and highlighting the importance of regulating the referral process through a standardised and rational approach.

To the best of our knowledge, this is the first formal assessment of the integration of SATS in the referral protocol of a low-income country. The decision to introduce the SATS to improve the existing referral service stemmed from its proven suitability for a low-resource context, evidenced by its algorithm-based approach, the availability of ready-to-use charts and training manuals, and the inclusion of easy-to-record vital signs [18,27,28].

Nevertheless, the use of SATS in this prehospital setting was associated with high rates of incorrect triage compared to the thresholds of 15% and 10% for over- and undertriage, respectively, which are usually deemed to be acceptable in hospital settings [24]. Our results are consistent with those of Mould-Millman and colleagues reporting high rates of undertriage among prehospital providers asked to assess clinical vignettes and assign a final SATS triage colour [24]. The authors suggested that the degree of miscalculation of TEWS, identified as one the most frequent causes of error in assigning the final triage colour could be further exacerbated under stressful real-life circumstances, such as the management of acute patients. The incorrect identification of predetermined emergency conditions and the incorrect selection of clinical discriminators are other common errors reported in the literature [24,29].

Furthermore, our analysis focused exclusively on under- and overtriage rates without investigating the root causes behind the decisions by nurses. The low inter-rater agreement and overall high rates of incorrect triage could have been amplified by the specific context. Indeed, the triage process is a complex activity, often performed in a stressful, distracting environment [30] that can influence the triage code assigned by nurses independently from the intrinsic reliability of the protocol [31]. This could, at least partially, justify the poorer SATS performance when compared to that in vignette-based studies [24].

This suggests the need for further studies to better assess the practical limitations for the implementation of SATS in a prehospital setting.

On the other hand, the main aim of the referral protocol implemented in the urban ambulance system of Beira was to avoid the transportation of green codes to HCB. The misclassification of yellow, orange, and red codes did not affect the referral indication. Among the 126 green-code patients erroneously referred to HCB, the vast majority were undertriaged (Table 3). This could mean that nurses could have recognised the mismatch between the assigned green code and the actual clinical condition, subsequently deciding to refer patients to HCB anyway, a frequent phenomenon already described in the literature [22,24,29]. Despite this procedure being contemplated in the SATS [32], it could have led to an apparent incorrect triage due to the contrast between recorded clinical parameters and triage codes assigned retrospectively.

Furthermore, the delivered training course might not have been sufficient in terms of content, duration, and teaching methodologies to provide sufficient skills and knowledge to the nurses. On the other hand, the available literature highlights a great heterogeneity among SATS training, thus indicating that no standardisation currently exists; for example, the duration of courses varies from 1-h [19,20] to 2-day courses [21]. In addition, the official SATS training manual [32] does not provide a suggested duration nor a specific teaching approach.

Moreover, this referral service represents, especially for the vulnerable strata of population, one of the few chances to reach HCB. Subsequently, nurses could have decided to refer patients with specific health needs even if not presenting an urgent condition recognised by SATS. Despite this approach overtaking the initial design and objectives of the referral system, it is very common and could be considered acceptable in the specific low-resource context [33]. In the same vein, the SATS has been implemented without any specific adaptation. Despite this approach being deemed necessary to provide a reliable and evidence-based tool, it does not take in consideration the specific needs and peculiarities of Beira’s prehospital context.

Hence, whether SATS represents the best available option to regulate acute patients’ referral in the prehospital setting is still under debate. Studies that assess the practical implications, benefits, and limitations of SATS implementation are needed to better understand its use in this context, and whether any adaptation is required to adjust the algorithm according to prehospital and local particularities.

Our findings should be interpreted considering some limitations. First, the study was performed immediately after the introduction of SATS, thus intercepting the learning curve of local nurses, and not investigating the long-term effects of the implementation. In addition, data used for this study were gathered from patients’ referral charts, thus only including information on transported patients. Therefore, we are unaware of the actual number of patients assessed and treated at each HC in the selected timeframe, and we could not estimate the proportion of the different codes assigned to patients who have not been referred to the HCB. Additionally, as available data do not provide the outcome after hospitalisation, we were unable to establish whether the prioritisation of severe cases in the referral system effectively translated into improved health outcomes.

## 5. Conclusions

The integration of SATS in the Beira urban ambulance referral system has the potential to regulate the transport of acute patients to the central hospital, thus increasing the number of severe cases receiving advanced medical care. Despite nurses’ performance being ameliorated with the routine application of the protocol, the use of SATS in this setting was associated with high rates of incorrect triage, suggesting a possible scope for improvement. For instance, the implementation could be strengthened by specific context-based tailoring, and the training might be designed and delivered taking into consideration the local professionals’ expertise and habits.

A similar approach can be replicated in analogous settings to further investigate factors that might impact SATS application in prehospital settings, and to explore possible strategies for protocol adaptation and training improvement.

## Figures and Tables

**Figure 1 ijerph-19-10298-f001:**
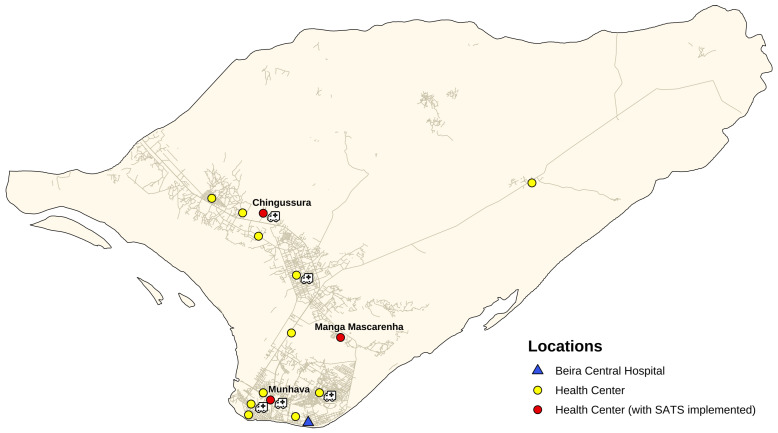
Distribution of health centres (HCs) and ambulances in Beira.

**Figure 2 ijerph-19-10298-f002:**
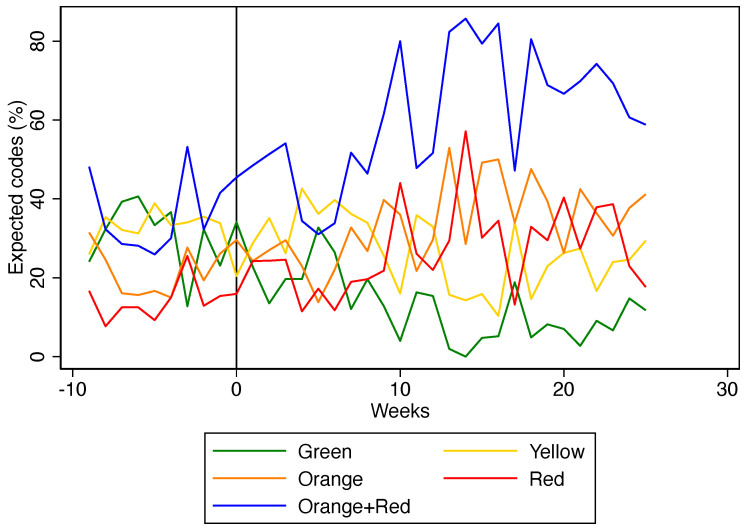
Proportion of expected triage codes before and after the implementation of the SATS (black line).

**Figure 3 ijerph-19-10298-f003:**
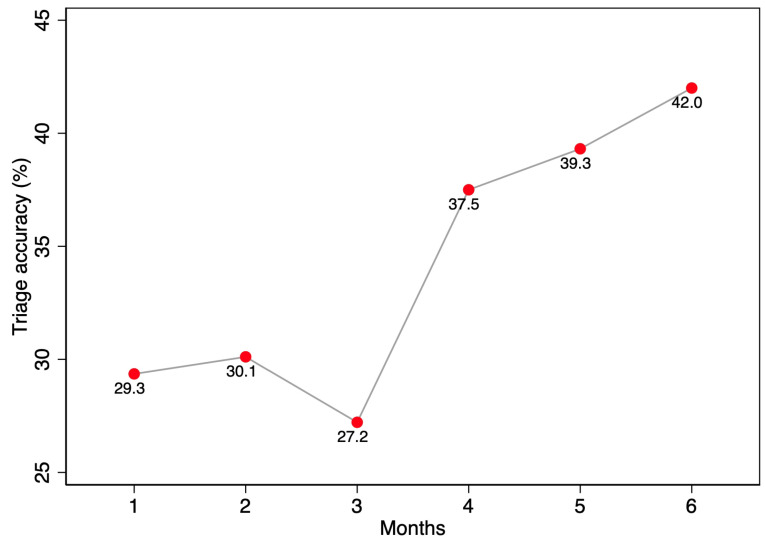
Triage accuracy (proportion of assigned codes matching with expected codes) in the postimplementation phase.

**Table 1 ijerph-19-10298-t001:** Demographic information of referred patients before and after SATS implementation.

	Preimplementation	Postimplementation
**Sex, n (%)**		
Male	171 (30.0%)	474 (29.5%)
Female	381 (70.0%)	1134 (70.5%)
Pregnant	183 (48.0%)	609 (53.7%)
**SATS age classes**		
0–2 years (younger child)	88 (15.9%)	270 (16.8%)
3–12 (older child)	68 (12.3%)	147 (9.1%)
12+ (adult)	396 (71.7%)	1191 (74.1%)
**Health centre**		
Chingussura	356 (64.5%)	827 (51.4%)
Manga Mascarenhas	79 (14.3%)	177 (11.0%)
Munhava	117 (21.2%)	604 (37.6%)
**Disease category**		
Medical	97 (17.6%)	332 (20.7%)
Obstetric/gynaecological	141 (25.5%)	652 (40.6%)
Paediatric	118 (21.4%)	422 (26.2%)
Surgical/trauma	57 (10.3%)	202 (12.6%)
Missing	139 (25.2%)	0 (0.0%)
Observation time (days)	61	176
Transport rate (patients/day)	9.1	9.1
Total	552	1608

**Table 2 ijerph-19-10298-t002:** Expected codes of referred patients before and after SATS implementation.

Code	Preimplementation	Postimplementation	*p*-Value
Green	174 (31.5%)	206 (12.8%)	
Yellow	181 (32.8%)	434 (27.0%)	
Orange	121 (21.3%)	539 (33.5%)	<0.0005
Red	76 (13.8%)	429 (26.7%)	
Total	552	1608	

**Table 3 ijerph-19-10298-t003:** Comparison of the triage code originally assigned by local personnel and the code retrospectively obtained by researchers. The results in italics represent the number of correct triages performed by local personnel; those above and below them identify the percentage of over- and undertriage per category, respectively.

		Expected				Undertriage	Overtriage	Total
		Green	Yellow	Orange	Red			
**Assigned**	**Green**	*29 (23.0%)*	67	25	5	97 (77.0%)	-	126
	**Yellow**	62	*142 (31.3%)*	173	79	252 (55.3%)	62 (13.6%)	456
	**Orange**	94	181	*268 (34.5%)*	235	235 (30.2%)	275 (35.4%)	778
	**Red**	21	44	73	*110 (44.4%)*	-	138 (55.7%)	248
	**Total**	206	434	539	429	584 (36.3%)	475 (29.5%)	1608

## Data Availability

The datasets used and/or analysed during the current study are available from the corresponding author on reasonable request.

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
