# Peer review of "Implementation of the South African Triage Scale (SATS) in a New Ambulance System in Beira, Mozambique: A Retrospective Observational Study"

_ijerph, 2022, doi:10.3390/ijerph191610298_

Round 1

Reviewer 1 Report

Work meaningful, readable few comments:

-please rewrite the topic

- please rewrite the abstract

- please rewrite the conclusion

Author Response

We thank the reviewer. We found it hard to interpret the request to rewrite parts of the paper without any specific indication on what issue to specifically address. Nonetheless, we have reviewed completely the whole manuscript to improve its readability.

Reviewer 2 Report

Very basic study that only shows the problem and provides poor results. They do not address the reasons why the application of the system has such a high error rate, nor do they provide an answer to this need.As the authors state, they only wanted to determine whether triage was correct or not. In this sense, the authors are aware of the deficient use of the triage protocol and blame the results on the characteristics of the countries in which it is applied and not, as might be thought, on a deficient application of the protocol. Perhaps greater attention to this casuistry could elucidate the reasons and thus elaborate a set of proposals for the improvement of the protocol (training of nursing personnel and adjustment/adaptation of the protocol to specific scenarios). As recognized by the authors, the study has many gaps and they are unaware of key parameters (number of patients accessing and not assigned to orange/red codes) to correctly establish the problem and its solution. As the study is set out, I suggest a thorough revision for publication.

Author Response

Dear Reviewer,
thank you for for the time spent reading and commenting the manuscript. Please find below our answers to the comments.

Very basic study that only shows the problem and provides poor results. They do not address the reasons why the application of the system has such a high error rate, nor do they provide an answer to this need.

This issue pointed out by the reviewer is highly relevant; however, it is worth considering that error rates reported in our study are consistent with those reported in the literature among pre-hospital providers (Mould-Millman, N.K.; Dixon, J.M.; Burkholder, T.; Pigoga, J.L.; Lee, M.; de Vries, S.; Moodley, K.; Meier, M.; Colborn, K.; Patel, C.; 317 et al. Validity and reliability of the South African Triage Scale in prehospital providers. BMC Emergency Medicine 2021, 21, 8. 318 https://doi.org/10.1186/s12873-021-00406-6.). The objective of the paper was to assess whether nurses working at three health centers in Beira could use the SATS tool to triage patients for transfer to the central hospital and how this approach changed rates of referral. The model defined that patients with a green code should have stayed at the health centre, while patients with a red, orange, or yellow code were deemed as suitable for transfer to Beira. As such, the only misclassification that affected the referral indication was the one related to green codes. In fact, misclassification between red, orange, and yellow did not have any effect on the referral indication. Post implementation there were a higher percentage of red and orange patients and a lower number of green patients suggesting that the concept of patient prioritization (previously not used in the referral system) was understood and applied by nurses. According to us, this could be interpreted as a promising output, as it shows that the “first-in-first-out” approach has been successfully replaced with a structured methodology.

Following the reviewer’s suggestion, in the discussion section we tried to elaborate a series of rational explanations of the incorrect referral of 126 green patients to the HCB, which include:

  • Mismatch between the triage code assigned and the actual clinical condition (a procedure that is contemplated by the algorithm)
  • Possibility that, despite being triaged as green, patients had clinical conditions that could not be treated at the peripheral level.

Moreover, we acknowledge the need to further analyze root causes behind the decision of nurses (including miscalculation of TEWS and incorrect identification of predetermined emergency conditions – which we could not evaluate and analyze due to lack of data).

As the authors state, they only wanted to determine whether triage was correct or not. In this sense, the authors are aware of the deficient use of the triage protocol and blame the results on the characteristics of the countries in which it is applied and not, as might be thought, on a deficient application of the protocol. Perhaps greater attention to this casuistry could elucidate the reasons and thus elaborate a set of proposals for the improvement of the protocol (training of nursing personnel and adjustment/adaptation of the protocol to specific scenarios).

We would like to clarify that the study is not assessing the accuracy of the SATS in identifying patients appropriate for transfer to the central hospital, but the accuracy in the application of the tool. In our study, we point out that accuracy in the application of the SATS increased with time. Despite the high rates of incorrect triage, rather than a “deficient use of the triage protocol”, this observation of progressive improvement is suggesting that nurses might need more time to master the SATS tool, and that it is possible that the tool was not completely embedded after six months. In our extensive research and field experience in LICs, we have gained a growing awareness about the role of the context when evaluating the impact of health interventions. The context is not to be “blamed”, rather is something that can clearly influence the success/failures of any health interventions. As such, we believe that it is not possible to disregard this aspect when analyzing factors that might have impact the application of the SATS in our setting. For this reason, it is important to underline that the main message of the paper is that more training would be desirable to improve the accuracy in the application of the tool. Nonetheless, following the reviewer suggestion, we recognize that this concept might need to be reinforced in the paper. As such, we have introduced a new paragraph in the discussion addressing this issue. Moreover, we agree with the reviewer that the adjustment/adaptation of the protocol to specific scenarios might be needed. Therefore, we have added this insightful aspect in the paper.

As recognized by the authors, the study has many gaps and they are unaware of key parameters (number of patients accessing and not assigned to orange/red codes) to correctly establish the problem and its solution. As the study is set out, I suggest a thorough revision for publication.

We recognize that a significant limitation of the paper is that our study only assessed those patients who were referred. Unfortunately, we have no data to assess patients who were not referred. Nonetheless, while acknowledging this limitation and the need to further studies in the prehospital setting, we strongly believe that, as this study represents the first formal assessment of the implementation of SATS in the prehospital referral protocol of a LIC, sharing our experience is key to supporting other countries in implementing similar approaches.

In conclusion, we have thoughtfully revised the manuscript, taking in consideration the comments provided by the reviewers. Furthermore, we improved the english language.

Reviewer 3 Report

Sorry, this paper does not belong to my academic field. The follows are the comments I provide from the perspective of thesis writing logic, which can also be ignored.

1. It is better to use scientific counting method.

2. The Enlish expression of this paper is recommended to be improved.

3. No major quantitative analysis, modeling and innovative contributions can be found from the paper in my opinion.

4. What is the ordinate for Fig. 2?

5. What is the use of Fig. 1?

Author Response

We thank the reviewer for the time spent reading and commenting the manuscript. Please find blue our answers.

It is better to use scientific counting method.

I understand the reviewer is suggesting using the scientific notation (e.g., 4×103 instead of 4000). Despite the MDPI instruction for Authors do not indicate when use it – leaving the choices to the Authors- I would like to point out that the results do not include values higher than two thousand. In this specific context, we believe that using the scientific notation does not provide any added value to the readability of the text, nor helps the interpretation of the results.

The Enlish expression of this paper is recommended to be improved.

We thank the reviewer. Following his/her suggestion, we have proofread the document.

No major quantitative analysis, modeling and innovative contributions can be found from the paper in my opinion.

The purpose of the study was to “ assess whether the implementation of the SATS varied the pattern of acute patients referred and evaluate the correct use of the SATS by the nurses”. Consequently, we used the chi-squared test to compare the proportion of codes before and after the implementation. Furthermore, we used logistic regression to investigate the possible association between nurses’ accuracy of each referral and gender, age, and time since implementation of SATS (please see the Statistical Analysis section).

Our study does not provide itself a new triage protocol, nor has the potentiality to assess its impact on patients’ outcome. Since it is a retrospective observational study, it is based on information collected for other purpose. Unfortunately, for a lack of available data, it was not possible to gather more information than the ones already presented and discussed in the manuscript.

However, as already presented in our manuscript, no previous formal assessment of SATS have been conducted in low- and middle-income countries. Under this view, our manuscript provides a novel contribution. In the same vein, we believe that this type of assessment, despite not providing innovative tools, are fundamental to build a reliable knowledge base that can help practitioners and stakeholders to decide which triage protocol implement, together with the consequents benefits and limitations.

What is the ordinate for Fig. 2?

The caption reports the significance of the figure (“Proportion of expected triage codes before and after the implementation of the SATS”). However, we agree that could be difficult to the reader understanding it at first glance. Therefore, we added the label also on Y axis.

What is the use of Fig. 1?

Figure 1 shows the distribution of health centers among Beira city, highlighting the three involved in the study (red circles) between the others (yellow circles). Similarly, it shows the location of the hospital and of the ambulances.  Despite this figure is not presenting any result, we think it helps the reader to better understand the context and the challenges. In this type of studies, where the results and their significance should be evaluated considering the setting, a map is a useful tool to support the readers. Please see also this paper, in which we used the same approach: Caviglia M, Putoto G, Conti A, et al. Association between ambulance prehospital time and maternal and perinatal outcomes in Sierra Leone: a countrywide study. BMJ Glob Health. 2021;6(11):e007315. doi:10.1136/bmjgh-2021-007315

Round 2

Reviewer 2 Report

I thank the authors for taking my suggestions into consideration and for their approach and transfer of them to the study.

Thus, I consider the new contributions of the authors to be appropriate and suggest that these new appreciations have greater weight in the conclusions.

Perhaps a greater emphasis on them would give greater projection to their study. e.g.:

"On the other hand, the available literature highlights a great heterogeneity SATS training, thus indicating that no standardization currently exists; for example, the duration of courses varies from one hour [19,20] to two-days courses [21]. In addition, the official SATS Training Manual [30] does not provide a suggested duration nor a specific teaching approach."

Author Response

Dear Reviewer,
many thanks for the effort in reviewing our manuscript. Following your suggestion, we have improved the conclusions.